# Novel predictive factors for patient discomfort and severe cough during bronchoscopy: A prospective questionnaire analysis

**Fumi Kobayashi, Takeshi Saraya** **\*, Miku Oda, Sho Sakuma, Masato Watanabe, Saori Takata, Masaki Tamura, Hiroki Takakura, Keitaro Nakamoto, Kojiro Honda, Kosuke Ohkuma, Sunao Mikura, Manami Inoue, Aya Hirata, Nozomi Kurokawa, Tatsuya Shirai, Kaori Aso, Akinari Noda, Chika Miyaoka, Yuki Yoshida, Narishige Ishikawa, Kikuko Morita, Eriko Ieki, Hiromi Nakajima, Haruyuki Ishii, Hajime Takizawa**

Department of Respiratory Medicine, Kyorin University School of Medicine, Mitaka, Tokyo, Japan

\* sara@yd5.so-net.ne.jp

## Abstract

During bronchoscopy, discomfort is mainly caused by an unavoidable cough; however, there are no reports of any predictive factors for strong cough during bronchoscopy identified before the procedure. To clarify the factors underlying the discomfort status and predictive factors for strong cough during bronchoscopy, we prospectively evaluated patients who underwent bronchoscopy at Kyorin University Hospital between March 2018 and July 2019. Before and after bronchoscopy, the enrolled patients answered a questionnaire regarding the procedure. At the same time, bronchoscopists evaluated cough severity using a four-grade cough scale. We evaluated patient characteristics and predictive factors associated with bronchoscopy from the perspective of discomfort and strong cough. A total of 172 patients were ultimately enrolled in this study. On multivariate logistic regression analysis, comparison of the subjective data between the discomfort and comfort groups revealed that factors that were more common in the former group were younger age (OR = 0.96, $p$ = 0.002), less experienced bronchoscopist (OR = 2.08, $p$ = 0.047), and elevation of cough score per 1 point (OR = 1.69, $p$ < 0.001). Furthermore, the predictive factors for strong cough prior to performing bronchoscopy were female sex (OR = 2.57, $p$ = 0.009), EBUS-TBNA (OR = 2.95, $p$ = 0.004), and prolonged examination time of more than 36 min (OR = 2.32, $p$ = 0.022). Regarding patients' discomfort, younger age, less experienced bronchoscopist, and the elevation of cough score per 1 point were important factors for discomfort in bronchoscopy. On the other hand, female sex, EBUS-TBNA, and prolonged examination time were crucial factors for strong cough.

## Introduction

Flexible bronchoscopy is a pivotal tool for diagnosing diverse respiratory diseases, such as lung cancer, idiopathic or secondary interstitial pneumonias, and infectious lung diseases. In the

**Funding:** The author(s) received no specific funding for this work.

**Competing interests:** The authors have declared that no competing interests exist.

modern era of evolutionarily developed personalized medicine for the management of lung cancer, it is mandatory to perform tissue biopsy and next-generation sequencing [1] and examine tumor PD-L1 expression [2]. However, bronchoscopy is widely recognized as a procedure in which discomfort is mainly caused by unavoidable cough [3–4]. Furthermore, there are no reports on the predictive factors for strong cough in bronchoscopy identified before the procedure. We conducted a prospective study to clarify factors that affect patients' discomfort and cough severity during bronchoscopy using a questionnaire before and after performing the procedure.

## Aim

To clarify the factors leading to discomfort and to determine the predictive factors for the occurrence of strong cough during bronchoscopy.

## Materials and methods

### Patients

We prospectively studied consecutive patients who underwent bronchoscopy at Kyorin University Hospital (a 1,100-bed tertiary center in Tokyo) in both inpatient and outpatient settings from March 2018 to July 2019. We enrolled adult patients aged over 18 years with no cognitive disorders who provided written informed consent.

### Questionnaire and cough scores

**Evaluation of comfort or discomfort based on the questionnaire after bronchoscopy.** We administered questionnaires before and after bronchoscopy. The first questionnaire was administered before bronchoscopy (Table 1). The severity of anxiety was evaluated using a visual analog scale (VAS) ranging from 1 to 5, with a score of 1 representing the most positive outcome and 5 indicating the most negative outcome. After the patients recovered completely from the sedation for bronchoscopy, they responded to the six items of the second questionnaire (Table 1) based on a VAS, which was created based on a previously used questionnaire [5]. A VAS score of 1 meant no anxiety at all, no discomfort during pharyngeal anesthesia, or during or after bronchoscopy at all, and the examination time for bronchoscopy was very short or easily acceptable for performing repeat bronchoscopy. Furthermore, we specifically focused on the response to "Did you feel uncomfortable during bronchoscopy?" in the second questionnaire and divided all patients into two groups, which were defined as the comfort

**Table 1. Questionnaire.**

| Questionnaire before bronchoscopy | |
|---|---|
| 1. Is this the first time you have had a bronchoscopy? | Yes/No |
| 2. How about your anxiety before the procedure? | VAS score 1 to 5 |
| **Questionnaire after bronchoscopy** | |
| 1. How about discomfort in oropharyngeal anesthesia? | VAS score 1 to 5 |
| 2. Do you remember the detail of bronchoscopy? | Yes/No |
| 3. Can you feel uncomfortable during bronchoscopy? | VAS score 1 to 5 |
| 4. Can you feel uncomfortable after bronchoscopy? | VAS score 1 to 5 |
| 5. How about the duration of examination? | VAS score 1 to 5 |
| 6. Repeat bronchoscopy would be acceptable for you? | VAS score 1 to 5 |

VAS: Visual analog scale.

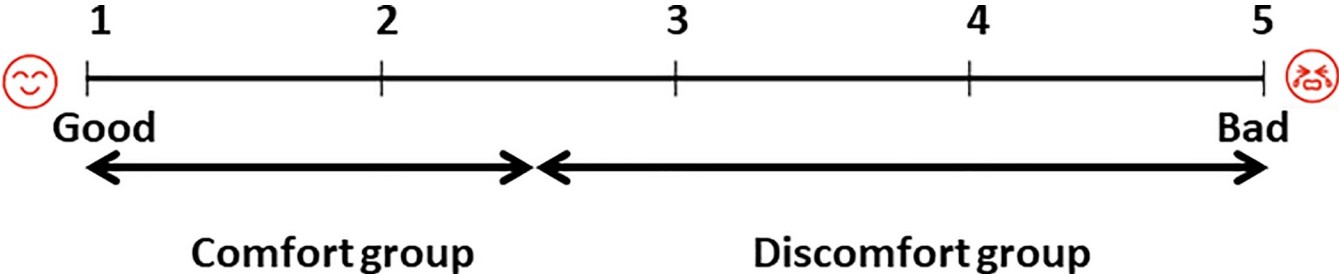

**Fig 1. Questionnaire-based visual analog scale for satisfaction during bronchoscopy.** Comfort group, VAS score 1 and 2; discomfort group, VAS score 3 to 5.

group (VAS score was 1 or 2) and discomfort group (VAS score was 3 to 5), and compared the two groups (Fig 1).

**Cough scores evaluated by physicians.** Soon after performing the bronchoscopy, before checking the results of the patients' questionnaire, the bronchoscopists and their assistants discussed and reached a consensus about the severity of cough during the procedure and divided patients into four grades, ranging from 0 to 3, which were defined as follows: 0, no cough; 1, slight cough:go ahead the procedure; 2, moderate cough: transient interruption of the procedure in the trachea; and 3, severe cough: removal of the bronchoscope from the trachea (Table 2). Thereafter, we classified the patients into strong (cough score = 2 or 3) and weak groups (cough score = 0 or 1) according to their cough score and compared the two groups. A schematic diagram of this study is shown in Fig 2. This study was approved by the Ethical Committee of Kyorin University (approval number: H29-174).

## Bronchoscopy procedure

All bronchoscopy procedures were performed using flexible bronchoscopes (BF-P290F, BF-1TQ290, or BF-UC260-OL8; Olympus, Tokyo, Japan). The procedures were performed in either an inpatient or an outpatient setting. Soon after arrival to the examination room, vital signs were evaluated. All patients were anesthetized by instillation of lidocaine (5 mL of 2%) into the throat with a Jackson-type spray. Midazolam for sedation and/or pethidine for analgesia was administered intravenously before bronchoscopy, and additional doses of midazolam were used to maintain moderate sedation. The initial use and additional dosages of midazolam and pethidine were at the discretion of the primary bronchoscopist. Some patients on dialysis did not receive pethidine. The scope was introduced through the mouth. When the bronchoscope was passed through the larynx and trachea or coughing occurred during the procedure, 2% lidocaine was administered through the bronchoscopy channel. Vital signs were monitored every 5 min in all patients. If oxygen saturation decreased, supplemental oxygen was administered to maintain 90% oxygen saturation, as measured using a pulse oximeter. After the procedure, 0.2 mg of flumazenil was administered intravenously and another 0.2 mg was co-

**Table 2. Cough score.**

| Score | Cough severity |
|:---:|---|
| 0 | No cough |
| 1 | A slight cough:go ahead the procedure |
| 2 | Moderate cough: transient interruption of procedure in the trachea |
| 3 | Severe cough: removal of the bronchoscope from the trachea |

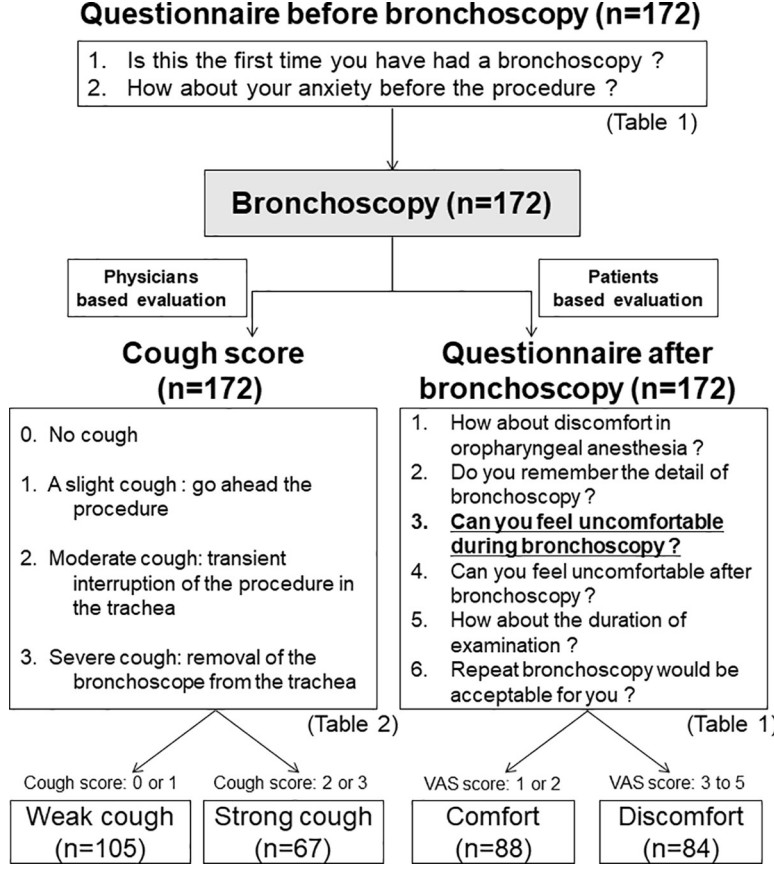

**Fig 2. Schema of this study.**

injected as an antagonist when midazolam was used. When only pethidine was used, 0.2 mg of naloxone was injected intravenously as an antagonist. Bronchoscopies were performed by pulmonologists whose experience ranged from 1 year or less to 18 years.

### Collection of associated data

The following data were retrieved from the patients: age, sex, weight, body mass index (BMI), bronchoscopy indications (lung tumor, infection, diffuse interstitial lung disease, sarcoidosis, and others), bronchoscopy examination time; the time from insertion of the tip of the bronchoscope into the oral cavity to removal from the mouth, method, and dose of sedative medications (pethidine alone, midazolam alone, or midazolam and pethidine), and type of bronchoscopy procedure (e.g., bronchoalveolar lavage [BAL], endobronchial ultrasound transbronchial biopsy with guide-sheath [EBUS-GS-TBB], endobronchial ultrasound-guided transbronchial needle aspiration [EBUS-TBNA], transbronchial biopsy [TBB], endobronchial biopsy [EBB], transbronchial lung biopsy [TBLB], brushing, observation, and treatment such as bronchothermoplasty [BT], insertion of airway stent, or endobronchial Watanabe spigot [EWS]). If two or more procedures were performed at the same time, both were counted. Among all enrolled cases, more than half of them underwent bronchoscopy by two or three bronchoscopists. In such cases, bronchoscopist experience was recorded as the one who mainly performed the procedure. The physician who has been engaged in bronchoscopy for

less than 3 years was defined as "less-experienced" bronchoscopist. The difference in vital signs before (baseline condition) and during the procedures (maximum values) were analyzed.

## Statistical analyses

Continuous variables are expressed as the mean ± standard deviation, unless otherwise stated, and were compared using a Mann–Whitney U test or t-test, as appropriate. Categorical variables were compared using the $\chi^2$ test or Fisher's exact test, as appropriate. The VAS score and cough score were treated as continuous variables. Multivariable analyses were performed using multiple logistic regressions. Univariate and multivariate analyses were used to identify the set of variables that could be used to classify the patients in the following aspects: discomfort vs. comfort and weak cough vs. strong cough.

Regarding the ROC curve for examination time, the cutoff point was determined as the minimum value of $[(1\text{-}sensitivity)^2 + (1\text{-}specificity)^2]$. $P < 0.05$ was considered to indicate a statistically significant difference. All statistical analyses were performed using EZR ver. 1.40 [6].

## Results

During the study registration period, 493 patients underwent planned bronchoscopies and 187 patients agreed to participate in the study. However, 15 patients were excluded from this study due to a lack of data. Finally, 172 patients were enrolled in this study.

## Examination time for each procedure

Participants were grouped depending on the procedure they underwent as follows: group 1 (EBUS-GS-TBB, n = 59), group 2 (EBUS-TBNA, n = 18), group 3 (BAL, n = 8), group 4 (TBLB, n = 4), group 5 (TBB, n = 3), group 6 (EBB, n = 6), group 7 (brushing, n = 9), group 8 (observation, n = 10), group 9 (BAL+TBLB, n = 14), group 10 (BAL+EBUS-TBNA, n = 5), group 11 (EBUS-TBNA+TBLB, n = 2), group 12 (EBB+EBUS-TBNA, n = 6), group 13 (GS-TBB+EBUS-TBNA, n = 16), group 14 (BAL+TBLB+EBUS-TBNA, n = 8), group 15 (Treatment [BT, airway stent, EWS], n = 4 [2, 1, 1, respectively]). The correlation between examination time and each procedure demonstrated that the procedure time in the EBUS-TBNA groups (groups 2 and 10–14; black column) was significantly longer than that in the non-EBUS-TBNA groups (groups 1, 3–9, 15; gray column) (41.3 ± 15.10 min vs. 33.4 ± 17.13 min, $p < 0.001$) (Fig 3A).

## Bronchoscopy comfort and discomfort group

Among all patients who underwent bronchoscopy (n = 172), the number in the discomfort and comfort groups was 84 and 88, respectively (Fig 2).

**Proportion of discomfort group in each procedure.**    Among all procedures, the proportion of patients with discomfort seemed to be greater in the EBUS-TBNA group than in the non-EBUS-TBNA group (Fig 3B), but the difference was not statistically significant (33/22 [60.0%] vs. 51/66[43.6%]; $p = 0.051$).

**Univariable analysis of discomfort and comfort groups.**    Comparison of discomfort and comfort groups revealed that younger age (64.06 ± 14.10 vs. 71.43 ± 11.77 years; $p < 0.001$), sarcoidosis as the indication for bronchoscopy (n = 10 [11.9%] vs. 3 [3.4%]; $p = 0.044$), BAL (n = 23 [27.4%] vs. n = 12 [13.6%]; $p = 0.041$), less-experienced bronchoscopist (n = 64 [76.2%] vs. n = 52 [59.1%]; $p = 0.024$), long examination time (41.96 ± 17.46 vs. 35.15 ± 16.36 min; $p = 0.009$), elevation of cough score per 1point (1.43 ± 1.04 vs. 0.73 ±0.93; $p < 0.001$), prominent discrepancy between baseline and maximum values in systolic blood pressure

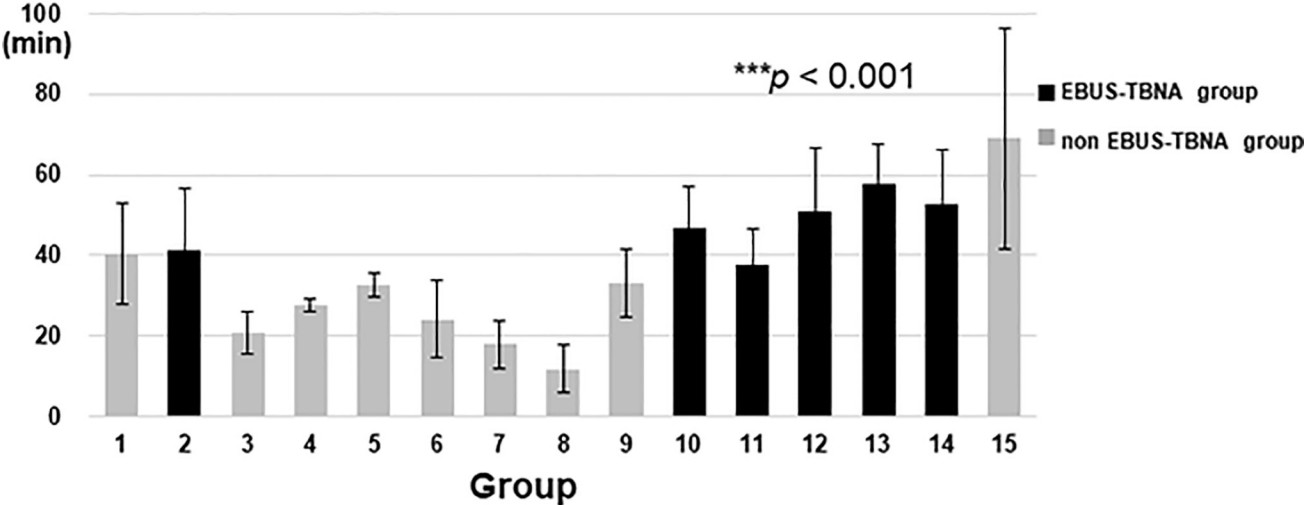

**A** Examination time in each procedure

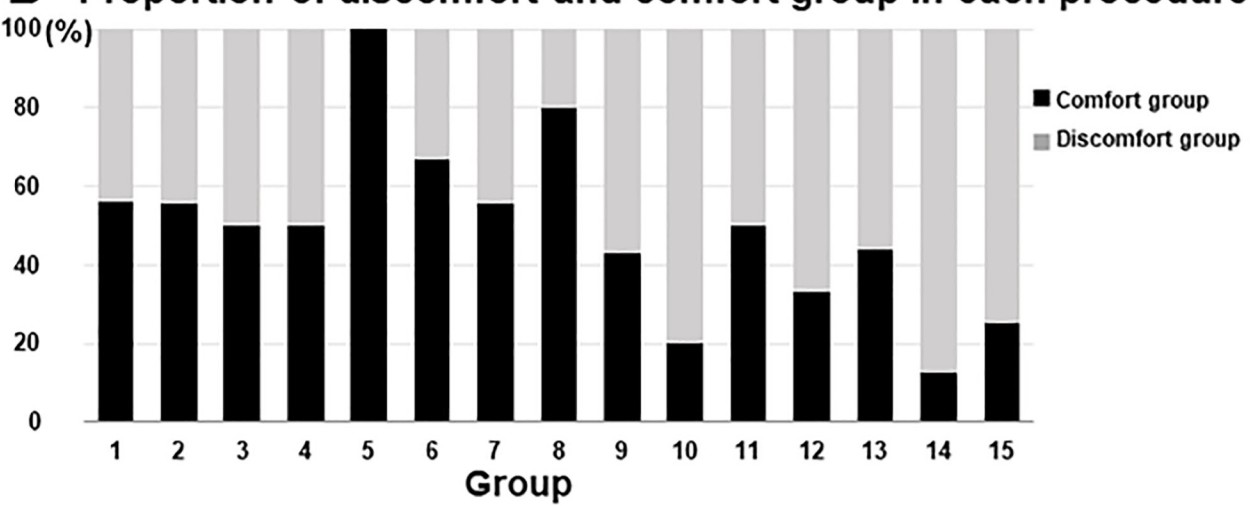

**B** Proportion of discomfort and comfort group in each procedure

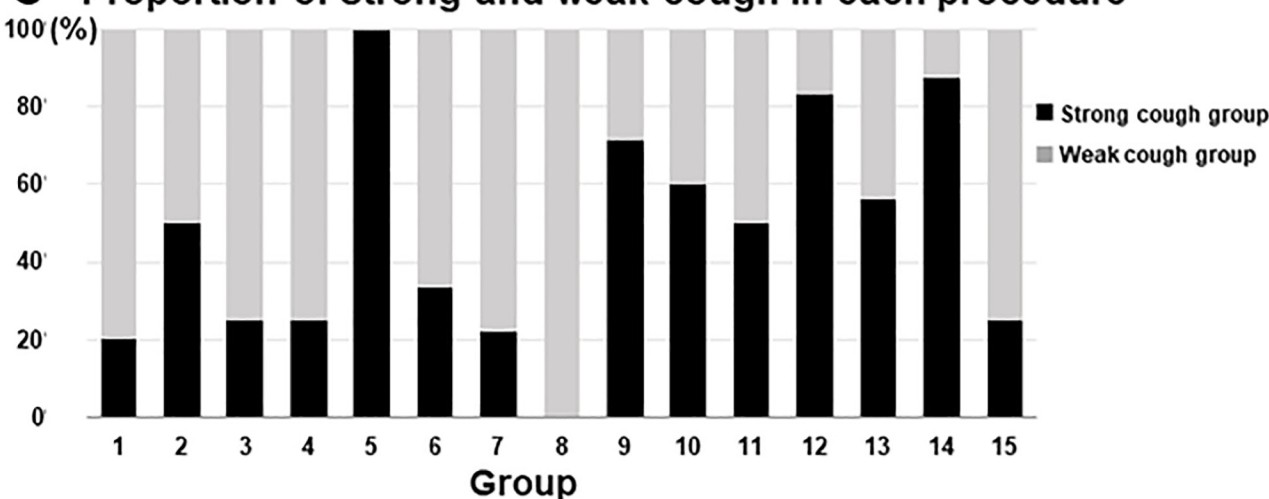

**C** Proportion of strong and weak cough in each procedure

**Fig 3. Bar charts showing the distribution of different factors across groups.** (A) Type of procedure with the required time split into EBUS-TBNA (black column) and non-EBUS-TBNA groups (grey column). The *p*-value indicates the results of the EBUS-TBNA and non-EBUS-TBNA group comparison. Each group was defined as follows: group 1 (EBUS-GS-TBB, n = 59), group 2 (EBUS-TBNA, n = 18), group 3 (BAL, n = 8), group 4 (TBLB, n = 4), group 5 (TBB, n = 3), group 6 (EBB, n = 6), group 7 (Brushing, n = 9), group 8 (Observation, n = 10), group 9 (BAL+ TBLB, n = 14), group 10 (BAL +EBUS-TBNA, n = 5) group 11 (EBUS-TBNA+TBLB, n = 2), group 12 (EBB+EBUS-TBNA, n = 6), group 13 (GS-TBB+EBUS-TBNA, n = 16), group 14 (BAL+TBLB+EBUS-TBNA, n = 8), and group 15 (Treatment [BT, airway stent, EWS] n = 4 [2, 1, 1, respectively]). (B) Proportion of comfort (black column) and discomfort patients (grey column) for each procedure. (C) Comparison of the strong cough (black column) and weak cough groups (grey column). BAL, bronchoalveolar lavage; EBB, endobronchial biopsy; EBUS-GS-TBB, endobronchial ultrasonography with a guide sheath transbronchial biopsy; EBUS-TBNA, endobronchial ultrasound-guided transbronchial needle aspiration; TBB, transbronchial biopsy; TBLB, transbronchial lung biopsy.

(25.64 ± 26.60 vs. 14.74 ± 26.70 mmHg; *p* = 0.004), heart rate (18.95 ± 19.23 vs. 11.26 ± 19.35; *p* = 0.006), and high anxiety score on the VAS scale (3.81 ± 1.15 vs. 3.28 ± 1.39; *p* = 0.016) were more common in the discomfort group (Table 3).

Based on the patient questionnaire administered after bronchoscopy, the discomfort group was more aware and felt the examination time was too long during bronchoscopy (Table 4).

**Multivariate analysis of the discomfort and comfort groups.** We performed logistic regression analysis using the significant variables in the univariate analysis, such as age, sarcoidosis as the indication for bronchoscopy, BAL, less-experienced bronchoscopist, examination time, cough score, Δ systolic BP (mmHg), Δ Heart rate (/min), and anxiety. Only three factors, younger age (odds ratio [OR] = 0.96; 95% confidence interval [CI]: 0.93–0.98; *p* = 0.002), less-experienced bronchoscopist (OR = 2.08; 95%CI: 1.01–4.28; *p* = 0.047), and elevation of cough score per 1point (OR = 1.89; 95% CI: 1.35–2.65; *p* < 0.001) (Table 5) were significantly associated with discomfort.

## Cough score evaluated by physicians

Among all patients who underwent bronchoscopy (n = 172), the number of patients in the strong cough and weak cough groups was 67 and 105, respectively (Fig 2).

**Proportion of patients with strong cough in each procedure.** The proportion of patients with strong cough (scores 2 and 3) was significantly greater in the EBUS-TBNA group than in the non-EBUS-TBNA group (34/21 [61.8%] vs. 33/84 [28.2%]; *p* < 0.001) (Fig 3C).

**Univariable analysis of weak cough and strong cough groups.** In the univariable analysis, predisposing factors for strong cough were female sex (n = 30 [44.8%] vs. n = 25 [23.8%]; *p* = 0.007), sarcoidosis as an indication for bronchoscopy (n = 10 [14.9%] vs. n = 3 [2.9%]; *p* = 0.006), BAL (n = 22 [32.8%] vs. n = 13 [12.4%]; *p* = 0.002), TBLB (n = 19 [28.4%] vs. n = 9 [8.6%]; *p* = 0.001), EBUS-TBNA (n = 34 [50.7%] vs. n = 21 [20.0%]; *p* < 0.001), non-performed EBUS-GS-TBB (n = 46 [68.7%] vs. n = 51 [48.7%]; *p* = 0.015), non-performed observation (n = 67 [100%] vs. n = 95 [90.5%]; *p* = 0.007) and examination time (44.58 ± 15.67 vs. 34.58 ± 17.07 min; *p* < 0.001). On further analysis, it was found that if the threshold of the examination time was set at 36 min, the diagnostic yield for strong cough had a sensitivity of 68.7%, specificity of 60.0%, and area under the curve of 0.68 (95%CI: 0.60–0.76). The proportion of patients with prolonged examination time (≥36 min) was also greater in the strong cough group (n = 46, 68.6%) than in the weak cough group (n = 42, 40.0%; *p* < 0.001) (Table 6).

**Multivariate analysis to identify predictors of strong cough groups.** Logistic regression analysis to identify predictors of the strong cough groups, among the variables that showed significance in the univariate analysis, female sex, sarcoidosis as an indication for bronchoscopy, BAL, TBLB, EBUS-TBNA, non-performed GS-TBB, non-performed observation and examination time (≥ 36 min), demonstrated independent associations between female sex (OR = 2.57; 95%CI: 1.27–5.21; *p* = 0.009), EBUS-TBNA (OR = 2.95; 95% CI: 1.40–6.18; *p* = 0.004), and examination time (≥36 min) (OR = 2.32; 95% CI: 1.13–4.76; *p* = 0.022) (Table 7).

**Table 3. Patient characteristics and univariate analysis for discomfort of bronchoscopy.**

| Factor | Total n = 172 | Discomfort n = 84 | Comfort n = 88 | *p value* |
|---|---|---|---|---|
| Age (year) [a] | 67.83 ± 13.44 | 64.06 ± 14.10 | 71.43 ±11.77 | < 0.001 |
| Gender (Male/Female), n (%)[d] | 117/55 (68.0/32.0) | 51/33 (60.7/39.3) | 66/22 (75.0/25.0) | 0.065 |
| Suspected disease | | | | |
| Lung tumor, n (%)[d] | 106 (61.6) | 48 (57.1) | 58 (65.9) | 0.305 |
| Interstitial pneumonia, n (%)[d] | 30 (17.4) | 16 (19.0) | 14 (15.9) | 0.733 |
| Sarcoidosis, n (%)[c] | 13 (7.6) | 10 (11.9) | 3 (3.4) | 0.044 |
| Infectious disease, n (%)[c] | 11 (6.4) | 4 (4.8) | 7 (8) | 0.536 |
| Other, n (%)[d] | 12 (7.0) | 6 (7.1) | 6 (6.8) | 1.000 |
| Weight (kg) [a] | 57.60 ± 11.52 | 56.71 ± 11.51 | 58.53 ± 11.53 | 0.628 |
| BMI (kg/m$^2$) [a] | 22.1 ± 3.35 | 22.43 ± 3.67 | 21.83 ± 3.00 | 0.565 |
| Outpatient, n (%)[c] | 8 (4.65) | 4 (4.8) | 4 (4.5) | 1.000 |
| Procedure | | | | |
| BAL, n (%)[d] | 35 (20.3) | 23 (27.4) | 12 (13.6) | 0.041 |
| TBLB, n (%)[d] | 28 (16.3) | 18 (21.4) | 10 (11.4) | 0.114 |
| EBUS-GS-TBB, n (%)[d] | 75 (43.6) | 35 (41.7) | 40 (45.5) | 0.730 |
| EBUS-TBNA, n (%)[d] | 55 (32.0) | 33 (39.3) | 22 (25.0) | 0.065 |
| EBB, n (%)[d] | 12 (7.0) | 6 (7.1) | 6 (6.8) | 1.000 |
| TBB, n (%)[c] | 3 (1.7) | 0 (0) | 3 (3.4) | 0.246 |
| Brushing, n (%)[c] | 9 (5.2) | 4 (4.8) | 5 (5.7) | 1.000 |
| Observation, n (%)[c] | 10 (5.8) | 2 (2.4) | 8 (9.1) | 0.100 |
| Treatment, n (%)[c] | 4 (2.3) | 3 (3.6) | 1 (1.1) | 0.359 |
| Less-experienced bronchoscopist, n (%) [d] | 116 (67.4) | 64 (76.2) | 52 (59.1) | 0.024 |
| Drug | | | | |
| Midazolam and pethidine, n (%)[d] | 159 (92.4) | 77 (91.7) | 82 (93.2) | 0.930 |
| Midazolam only, n (%)[c] | 2 (1.2) | 1 (1.2) | 1 (1.1) | 1.000 |
| Pethidine only, n (%)[c] | 11 (6.4) | 6 (7.1) | 5 (5.7) | 0.763 |
| Total midazolam (mg) [a] | 2.70 ± 1.58 | 2.95 ± 1.75 | 2.45 ± 2.36 | 0.050 |
| Total pethidine (mg) [a] | 34.59 ± 5.97 | 34.58 ± 6.06 | 34.60 ± 5.92 | 0.977 |
| Examination time (min) [b] | 38.48 ± 17.20 | 41.96 ± 17.46 | 35.15 ± 16.36 | 0.009 |
| Cough score [a] | 1.07 ± 3.35 | 1.43 ± 1.04 | 0.73 ±0.93 | < 0.001 |
| Δ systolic BP (mmHg) [a] | 20.06 ± 27.13 | 25.64 ± 26.60 | 14.74 ± 26.70 | 0.004 |
| Δ Heart rate (/min) [a] | 15.02 ± 19.62 | 18.95 ± 19.23 | 11.26 ± 19.35 | 0.006 |
| Questionnaire before bronchoscopy | | | | |
| Retry (1st time/≧ 2nd time), n (%)[d] | 135/37 (78.5/21.5) | 65/19 (77.4/22.6) | 70/18 (79.5/20.5) | 0.873 |
| Anxiety [a] | 3.54 ± 1.30 | 3.81 ± 1.15 | 3.28 ± 1.39 | 0.016 |

BAL, bronchoalveolar lavage; BMI, body mass index; BP, blood pressure; EBB, endobronchial biopsy; EBUS-GS-TBB, endobronchial ultrasonography with a guide sheath transbronchial biopsy, EBUS-TBNA: Endobronchial ultrasound-guided transbronchial needle aspiration, TBB: transbronchial biopsy, TBLB: transbronchial lung biopsy, Data are presented as mean ± standard deviation.

a: Mann–Whitney U test

b: t-test.

c: Fisher's exact test

d:$\chi^2$ test.

## Discussion

This study demonstrated that younger age, less-experienced bronchoscopist, and elevation of cough score per 1point were associated with discomfort. Importantly, with special reference to

**Table 4. Univariate analysis of questionnaire after bronchoscopy for discomfort of bronchoscopy.**

| Questionnaire after bronchoscopy | Total n = 172 | Discomfort n = 84 | Comfort n = 88 | p value |
|---|---|---|---|---|
| Oropharyngeal anesthesia [a] | 2.54 ± 1.32 | 2.00 ± 1.02 | 3.06 ± 1.38 | < 0.001 |
| Memory, n (%)[b] | 113 (65.7) | 64 (76.2) | 49 (55.7) | 0.008 |
| Feeling after examination [a] | 2.70 ± 1.00 | 3.11 ± 0.98 | 2.31 ± 0.86 | < 0.001 |
| Feeling about examination time [a] | 2.84 ± 0.86 | 3.26 ± 0.78 | 2.44 ± 0.83 | < 0.001 |
| Willing for repeat bronchoscopy [a] | 2.83 ± 1.23 | 3.36 ± 1.13 | 3.32 ± 1.12 | < 0.001 |

Data are presented as mean ± standard deviation.

a: Mann–Whitney U test

b: $\chi^2$ test.

the severity of cough assessed using a cough score, we provide the first evidence that female sex, EBUS-TBNA, and prolonged examination time (≥36 min) are predictive factors for strong cough in bronchoscopy.

Regarding discomfort during bronchoscopy, previous reports suggested that female sex, unexpected pain [7], severe anxiety before bronchoscopy, bronchoscopist's experience [3], nasal insertion, and examination time [4] affect patient discomfort. Among the predictive factors that were significant in our multivariate analysis, less-experienced bronchoscopists had already been published in a previous report [3], but younger age and cough severity have not been widely reported. Previous studies enrolled relative younger patients (i.e., aged 64.7 ± 12.4 years [mean ± standard deviation] [7], 59.9 ± 15.1 years [3], and 56.5 ± 13.7 years [4]) compared to those enrolled in ours (67.83 ± 13.44 years), which might emphasize that younger age itself can be a predictive factor for discomfort, especially in an aging society like that of Japan. Interestingly, this study demonstrated that "cough severity" was another predictive factor for discomfort, but the relationship between cough and patient discomfort has been scarcely reported. In this regard, the severity of cough was assessed by physicians after reaching a consensus, and the probability of being in the discomfort group increased (OR = 1.69) per 1point elevation of cough score.

Furthermore, predictive factors for strong cough included female sex (OR = 2.57), EBUS-TBNA (OR = 2.95), and prolonged examination time (≥36 min) (OR = 2.32). In general, women are more vulnerable to chronic cough than men in the clinic setting [8], while it is not known whether female sex itself can affect cough severity, even in the setting of bronchoscopy. Therefore, this study clearly demonstrated that female sex has a clear association with strong cough from the physician's viewpoint. We also discovered that EBUS-TBNA is another predictive factor for strong cough. Even when sedated, cough can occur in 93% of patients during bronchoscopy [9], which is probably exaggerated by direct attachment of the EBUS probe to the main tracheal lumen or bronchus and the longer time required for completion of the procedure than the other bronchial examinations, as was seen in our study. Of note, this study

**Table 5. Multivariate analysis for discomfort of bronchoscopy.**

|  | Odds ratio | 95%CI | p value |
|---|---|---|---|
| Age | 0.96 | 0.93–0.98 | 0.002 |
| Less-experienced bronchoscopist | 2.08 | 1.01–4.28 | 0.047 |
| Cough score per 1 point | 1.89 | 1.35–2.65 | < 0.001 |

95%CI: 95% confidence interval.

**Table 6. Univariate analysis for strong cough.**

| | Weak cough n = 105 | Strong cough n = 67 | *p* value |
|---|---|---|---|
| Age (year) [a] | 69.35 ± 12.17 | 65.45 ± 15.02 | 0.141 |
| Gender (Male/Female), n (%)[d] | 80/25 (76.2/23.8) | 37/30 (55.2/44.8) | 0.007 |
| Suspected disease | | | |
| Lung tumor, n (%)[d] | 68 (64.8) | 38 (56.7) | 0.370 |
| Interstitial pneumonia, n (%)[d] | 16 (15.2) | 14 (20.9) | 0.455 |
| Sarcoidosis, n (%)[c] | 3 (2.9) | 10 (14.9) | 0.006 |
| Infectious disease, n (%)[c] | 9 (8.6) | 2 (3) | 0.205 |
| Other, n (%)[c] | 9 (8.6) | 3 (4.5) | 0.371 |
| Weight (kg) [a] | 57.70 ± 11.97 | 57.43 ± 10.88 | 1.000 |
| BMI (kg/m$^2$) [a] | 21.83 ± 3.44 | 22.59 ± 3.17 | 0.097 |
| Outpatient, n (%)[c] | 6 (5.7) | 2 (3.0) | 0.485 |
| Procedure | | | |
| BAL, n (%)[d] | 13 (12.4) | 22 (32.8) | 0.002 |
| TBLB, n (%)[d] | 9 (8.6) | 19 (28.4) | 0.001 |
| EBUS-GS-TBB, n (%)[d] | 54 (51.4) | 21 (31.3) | 0.015 |
| EBUS-TBNA, n (%)[d] | 21 (20.0) | 34 (50.7) | < 0.001 |
| EBB, n (%)[c] | 5 (4.8) | 8 (11.9) | 0.136 |
| TBB, n (%)[c] | 0 (0) | 3 (4.5) | 0.058 |
| Brushing, n (%)[c] | 7 (6.7) | 2 (3.0) | 0.485 |
| Observation, n (%)[c] | 10 (9.5) | 0 (0) | 0.007 |
| Treatment, n (%)[c] | 3 (2.9) | 1 (1.5) | 1.000 |
| Less-experienced bronchoscopist, n (%)[d] | 66 (62.9) | 50 (74.6) | 0.131 |
| Drug | | | |
| Midazolam and pethidine, n (%)[c] | 96 (91.4) | 63 (94.0) | 0.769 |
| Midazolam only, n (%)[c] | 1 (1.0) | 1 (1.5) | 1.000 |
| Pethidine only, n (%)[c] | 8 (7.6) | 3 (4.5) | 0.532 |
| Total midazolam (mg) [a] | 2.10 ± 1.27 | 3.63 ± 1.56 | < 0.001 |
| Total pethidine (mg) [a] | 34.50 ± 5.67 | 34.74 ± 6.46 | 0.695 |
| Examination time (min) [a] | 34.58 ± 17.07 | 44.58 ± 15.67 | < 0.001 |
| Examination time (min) ≧ 36, n (%)[d] | 42 (40.0) | 46 (68.6) | < 0.001 |
| ΔSystolic BP (mmHg)[b] | 11.57 ± 22.97 | 33.37 ± 27.95 | < 0.001 |
| ΔHeart rate (/min)[a] | 8.79 ± 16.84 | 24.78 ± 19.80 | < 0.001 |
| Questionnaire before bronchoscopy | | | |
| Retry (1st time/≧ 2nd time), n (%)[d] | 83/22 (79.0/21.0) | 52/15 (77.6/22.4) | 0.974 |
| Anxiety [a] | 3.41 ± 1.30 | 3.75 ± 1.28 | 0.056 |

BAL, bronchoalveolar lavage; BMI, body mass index; BP, blood pressure; EBB, endobronchial biopsy; EBUS-GS-TBB, endobronchial ultrasonography with a guide sheath transbronchial biopsy, EBUS-TBNA: Endobronchial ultrasound-guided transbronchial needle aspiration, TBB: transbronchial biopsy, TBLB: transbronchial lung biopsy, Data presented as mean ± standard deviation.

a: Mann–Whitney U test

b: t-test.

c: Fisher's exact test

d: χ$^2$ test.

clearly demonstrated that examination time over 36 min can predict strong cough regardless of whether EBUS-TBNA is performed.

This study has some limitations in that it was conducted in a single center; therefore, a multicenter prospective study with more cases is required. Furthermore, evaluation of cough

**Table 7. Multivariate analysis for strong cough.**

|  | Odds ratio | 95%CI | p value |
|---|---|---|---|
| Female | 2.57 | 1.27–5.21 | 0.009 |
| EBUS-TBNA | 2.95 | 1.40–6.18 | 0.004 |
| Examination time ≧ 36min | 2.32 | 1.13–4.76 | 0.022 |

EBUS-TBNA, Endobronchial ultrasound-guided trasbronchial needle aspiration; 95%CI, 95% confidence interval.

severity only depended on the physicians' consensus but not on quantitative methods, such as counting the absolute number of coughs [10]. In addition, during EBUS-TBNA, bronchoscopists specifically focus on the position of the probe in the trachea, which might lead to overestimation of the severity of cough.

## Conclusions

The present study demonstrated that younger age, bronchoscopist experience, and cough were associated with discomfort during bronchoscopy, and female sex, EBUS-TBNA, and prolonged examination time can be predictive factors for strong cough.

## Supporting information

**S1 File.**
(XLSX)

## Author Contributions

**Data curation:** Takeshi Saraya.

**Investigation:** Fumi Kobayashi, Takeshi Saraya, Miku Oda, Sho Sakuma, Masato Watanabe, Saori Takata, Masaki Tamura, Hiroki Takakura, Keitaro Nakamoto, Kojiro Honda, Kosuke Ohkuma, Sunao Mikura, Manami Inoue, Aya Hirata, Nozomi Kurokawa, Tatsuya Shirai, Kaori Aso, Akinari Noda, Chika Miyaoka, Yuki Yoshida, Narishige Ishikawa, Kikuko Morita, Eriko Ieki, Hiromi Nakajima, Haruyuki Ishii, Hajime Takizawa.

**Writing – original draft:** Fumi Kobayashi, Takeshi Saraya.

**Writing – review & editing:** Takeshi Saraya.

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
