## [Decision Letter · Decision Letter 0]

15 Jun 2020

PONE-D-20-07274

Novel predictive factors for patient discomfort and severe cough during bronchoscopy: a prospective questionnaire analysis

PLOS ONE

Dear Dr. Saraya,

Thank you for submitting your manuscript to PLOS ONE. After careful consideration, we feel that it has merit but does not fully meet PLOS ONE’s publication criteria as it currently stands. Therefore, we invite you to submit a revised version of the manuscript that addresses the points raised during the review process.

Thank you for giving me a chance to be an editor of your paper. According to the reviewers’ comments, please revise your manuscript as much as possible. I believe the comments from reviewers would encourage you to refine your paper.

We look forward to receiving your revised manuscript.

Kind regards,

Atsushi Miyamoto

Academic Editor

PLOS ONE

Journal Requirements:

2. Please provide additional details regarding participant consent. In the ethics statement in the Methods and online submission information, please ensure that you have specified whether consent was written or verbal/oral. If consent was verbal/oral, please specify: 1) whether the ethics committee approved the verbal/oral consent procedure, 2) why written consent could not be obtained, and 3) how verbal/oral consent was recorded. If your study included minors, please state whether you obtained consent from parents or guardians in these cases.

3. Please include additional information regarding the survey or questionnaire used in the study and ensure that you have provided sufficient details that others could replicate the analyses. If you developed and/or translated a questionnaire as part of this study and it is not under a copyright license more restrictive than Creative Commons Attribution (CC-BY), please include a copy, in both the original language and English, as Supporting Information.

4. Please amend the manuscript submission data (via Edit Submission) to include author Syo Sakuma.

5. Please amend your authorship list in your manuscript file to include author Sho Sakuma.

Reviewers' comments:

Reviewer's Responses to Questions

**Comments to the Author**

1. Is the manuscript technically sound, and do the data support the conclusions?

Reviewer #1: Yes

Reviewer #2: Partly

Reviewer #3: Partly

2. Has the statistical analysis been performed appropriately and rigorously? 

Reviewer #1: Yes

Reviewer #2: Yes

Reviewer #3: Yes

3. Have the authors made all data underlying the findings in their manuscript fully available?

Reviewer #1: Yes

Reviewer #2: Yes

Reviewer #3: No

4. Is the manuscript presented in an intelligible fashion and written in standard English?

Reviewer #1: Yes

Reviewer #2: Yes

Reviewer #3: Yes

5. Review Comments to the Author

Reviewer #1: In this paper the authors studied risk factors of patient discomfort and severe cough during bronchoscopy using data from 172 patient who underwent bronchoscopy at Kyorin University Hospital.

My questions/comments are as below.

It looks like “subjective analysis” was used to refer to analysis with discomfort as the endpoint and “objective analysis” was used to refer to analysis with “strong cough” as the endpoint. I found the use of subjective and objective confusing. Probably better to describe the endpoints directly.

It’s mentioned in the statistical analysis section that Mann-Whitney U test or t-test was used to compare continuous variable. Please label in the relevant tables which test was used for each continuous variable.

What were the covariates included in the logistic regression models? Were variables with VAS scores included as continuous variables?

I’d recommend showing continuous variable as the mean and standard deviation since medians can be the same for two groups while the p-value is still significant (e.g. Table 4.w Anxiety).

How was “36 minutes” determined to be the cutoff for prolonged examination time?

What does VAS 1-5 mean for question 5 (How about the duration of the examination?)? 1 means too long? Just right? uncomfortable length?

Are data on the bronchoscopists who performed the procedures available? Presumably, they can influence the comfort levels and maybe the severity of cough.

Figure 3. It is interesting that 100% participants in group 4 had strong cough while they were all in the comfort group. Any explanation for this?

Reviewer #2: Comments to the Author

While it is widely recognized that severe cough cause patients’ discomfort during bronchoscopy, predictive factors for severe cough have not been reported. The present prospective study reported factors associated with patients’ discomfort and cough severity during bronchoscopy.

Major points

The setting of the present prospective study has several problems.

1. The authors grouped patients into discomfort group and comfort group by use of questionnaire after bronchoscopy. The authors described that the discomfort group included patients who answered VAS score 1 or 2 to the question about discomfort during bronchoscopy and the comfort group included those who answered VAS score 3 to 5, in the materials and methods session, page 6 line82-82. However, they described that the comfort group included VAS score 1 or 2 and discomfort group included VAS score 3 to 5 at Figure 1. These were inconsistent?

2. The authors grouped patients into strong cough group and weak cough group by use of cough score, which was the original score defined by the authors and assessed by bronchoscopists. The strong cough group included patients whose cough score was 0 or 1, and the weak cough group 2 or 3. However, the difference between cough score 1 (slight cough) and 2 (moderate cough, no need to interrupt procedure) was ill defined. The authors defined the cough score as objective score, but it depended on bronchoscopists’ subjectivity.

3. The present study reported examination time was one of the factors which affected severe cough. Additionally, it is described that the procedure time in EBUS-TBNA groups was significantly longer than that in non-EBUS-TBNA-groups. The EBUS-TBNA groups included group 10-14, but they did not include group 2 (EBUS-TBNA). Why did the authors sort group 2 into the non-EBUS-TBNA-group? It was likely that the examination time did not be depend on EBUS-TBNA, but depended on using two procedures.

Minor points

1. The present study showed that patients in comfort group received more midazolam than those in discomfort group. The deep sedation could be important to comfort during bronchoscopy. In this study, there were patients who did not receive pethidine. How did the authors decide which they administered　 pethidine patients?

2. The authors described that patients’ discomfort reported to be associated with bronchoscopists’ experience in the discussion session. In this study, bronchoscopy was performed by pulmonologists whose experience ranged from 1 to 18 years. How did bronchoscopists’ experience affect the outcome of this study?

Reviewer #3: This study is very interesting. However some of detail is lacked.

First, how did you measured procedural time ? Is it from vocal code passing ? Is it to the end of procedure ? Please show me how.

Second, the procedural time is too long comparing with standardized procedure, especially for EBUS-TBNA group, and this result may be a confounding factor.

Third, Please tell me the use or not use of your additional lidocaine spraying. If not, you can reduce the cough level during procedure. And you'd better show the average dose of Midazolam and Petizine. Previously published report show that the usage of sedative drugs reduce the cough during BFS procedure. You should discuss about the appropriation of your sedation.

6. PLOS authors have the option to publish the peer review history of their article (what does this mean?). If published, this will include your full peer review and any attached files.

Reviewer #1: No

Reviewer #2: No

Reviewer #3: Yes: Takashi Niwa

---

## [Author Response · Author response to Decision Letter 0]

4 Aug 2020

[Date of submission] July, 24 2020

[Editor’s name] Atushi Miyamoto

PLOS ONE

Dear Editor: 

We wish to re-submit the manuscript titled “Novel predictive factors for patient discomfort and severe cough during bronchoscopy: a prospective questionnaire analysis”. The manuscript ID is PONE-D-20-07274

We thank you and the reviewers for your thoughtful suggestions and insights. The manuscript has benefited from these insightful suggestions. I look forward to working with you and the reviewers to move this manuscript closer to publication in PLOS ONE.

The manuscript has been rechecked and the necessary changes have been made as per the reviewers’ suggestions. A point-by-point response to all comments has been prepared and given below. 

Reviewers’ Comments to the Author

Reviewer #1: In this paper the authors studied risk factors of patient discomfort and severe cough during bronchoscopy using data from 172 patient who underwent bronchoscopy at Kyorin University Hospital.

My questions/comments are as below.

It looks like “subjective analysis” was used to refer to analysis with discomfort as the endpoint and “objective analysis” was used to refer to analysis with “strong cough” as the endpoint. I found the use of subjective and objective confusing. Probably better to describe the endpoints directly.

Response: Thank you for your suggestion. As for Fig 2, we revised the term from “subjective score” to “patient-based evaluation” as well as from “objective score” to “physician-based evaluation”. 

We also revised the heading from “Subjective scores on the comfort of bronchoscopy evaluated by patients” to “Evaluation of the comfort or discomfort based on the questionnaire after bronchoscopy” as well as from “Objective score of cough severity on bronchoscopy as evaluated by physicians” to “Cough score evaluated by physicians”

It’s mentioned in the statistical analysis section that Mann-Whitney U test or t-test was used to compare continuous variable. Please label in the relevant tables which test was used for each continuous variable.

Response: Thank you for your comments. We labeled the data in Tables 3, 4, and 6. (Pages 13–14 and 16–17, Lines 201–205, 208–210 and 241–245)

Comment: What were the covariates included in the logistic regression models? Were variables with VAS scores included as continuous variables?

Response: Thank you for these important comments.

In the view of analysis, the cough score and VAS score of anxiety were included as continuous variables. We also added the following sentences:

Regarding discomfort, 

We performed logistic regression analysis using the significant variables in the univariate analysis, such as age, sarcoidosis as the indication for bronchoscopy, BAL, less-experienced bronchoscopy, examination time, cough score, Δsystolic BP (mmHg), Δ Heart rate (/min), and VAS score of anxiety. Only three factors, younger age (odds ratio [OR] = 0.96; 95% confidence interval [CI]: 0.93–0.98; p < 0.01) less experienced bronchoscopy (OR = 2.08; 95%CI: 1.01–4.28; p < 0.05) and severe cough (OR = 1.69; 95% CI: 1.40–2.74; p < 0.001) (Table 5) were significantly associated with discomfort. (Pages 14–15, lines 212–218)

Similarly, regarding strong cough,

Logistic regression analysis to identify predictors of the strong cough groups, among the variables that showed significance in the univariate analysis, female sex, sarcoidosis as an indication for bronchoscopy, BAL, TBLB, EBUS-TBNA, non-performed GS-TBB, non-performed observation, and examination time (≥ 36 min), demonstrated independent associations between female sex (OR = 2.57; 95%CI: 1.27–5.21; p < 0.01), EBUS-TBNA (OR = 2.95; 95% CI: 1.40–6.18; p < 0.01), and examination time (≥36 min) (OR = 2.32; 95% CI: 1.13–4.76; p < 0.05). (Pages 17–18, line 247–253). 

These covariates are all categorical variables.

I’d recommend showing continuous variable as the mean and standard deviation since medians can be the same for two groups while the p-value is still significant (e.g. Table 4.w Anxiety).

Response: Thank you for your important suggestions. We revised the data from the median (range) to mean ± SD in Tables 3, 4, and 6. (Pages 13–14 and 16–17, Lines 201–205, 208–210 and 241–245)

How was “36 minutes” determined to be the cutoff for prolonged examination time?

Response: Thank you for your comment. We added these sentences to the statistical analyses section. 

Regarding the ROC curve for examination time, the cutoff point was determined as the minimum value of [(1-sensitivity) 2 + (1-specificity) 2]. (Page 9, lines 143–144)

What does VAS 1-5 mean for question 5 (How about the duration of the examination?)? 1 means too long? Just right? uncomfortable length?

Response: Thank you for your comments. We have added these sentences as follows:

A VAS score of 1 meant no anxiety at all, no discomfort during pharyngeal anesthesia or during and/or after bronchoscopy, and the examination time for bronchoscopy was very short or easily acceptable for performing repeat bronchoscopy. （Page 5, lines 71–73）

Are data on the bronchoscopists who performed the procedures available? Presumably, they can influence the comfort levels and maybe the severity of cough.

Response: Thank you for your valuable comment. As you pointed out, we re-analyzed the data for the bronchoscopists. In the first edition, we recorded the youngest physician when more than one bronchoscopist performed the bronchoscopy, but we decided to use the data of the bronchoscopist who performed the procedure the longest. Therefore, the results of the analysis for discomfort were highly variable, and both univariate and multivariate analyses showed that the experience of bronchoscopists had a significant impact on the patients’ discomfort. This means less-experienced bronchoscopists (less than 3 years) were identified as additional predictive factors for discomfort.

On the other hand, the bronchoscopist’s experience did not affect the cough during bronchoscopy.

We further added the following sentences in the material methods section; “Collection of the associated data” in the cases who were handled by more than one bronchoscopist. 

Among all enrolled cases, more than half of them underwent bronchoscopy by two or three bronchoscopists. In such cases, the bronchoscopist experience was recorded as the one who mainly performed the procedure. (Page 8, lines 129 -131) 

Figure 3. It is interesting that 100% participants in group 4 had strong cough while they were all in the comfort group. Any explanation for this?

Response: Thank you for your comments. In the original article, Groups 4 and 5, were TBLB (n=4) and TBB (n=3). However, we mistakenly labeled Fig 3. Therefore, we revised Fig 3, and group 4 (renamed as group 5) as TBB (n=3). (Page 10, line 154) We cannot precisely explain the reason for this dissociation; however, they were all elderly patients (mean age of 73.3 years), and they had some determining factors for the comfortable group by univariate analysis such as short examination time (average: 32.6 minutes), no BALF, and no suspicion of sarcoidosis. On the other hand, all procedures were performed by less-experienced bronchoscopists.

Reviewer #2: Comments to the Author

While it is widely recognized that severe cough cause patients’ discomfort during bronchoscopy, predictive factors for severe cough have not been reported. The present prospective study reported factors associated with patients’ discomfort and cough severity during bronchoscopy.

Major points

The setting of the present prospective study has several problems.

1. The authors grouped patients into discomfort group and comfort group by use of questionnaire after bronchoscopy. The authors described that the discomfort group included patients who answered VAS score 1 or 2 to the question about discomfort during bronchoscopy and the comfort group included those who answered VAS score 3 to 5, in the materials and methods session, page 6 line82-82. However, they described that the comfort group included VAS score 1 or 2 and discomfort group included VAS score 3 to 5 at Figure 1. These were inconsistent?

Response: Thank you for your valuable comments. We have revised this as follows.

…, which were defined as the comfort group (VAS score was 1 or 2) and discomfort group (VAS score was 3 to 5), and compared the two groups (Fig 1). (Page 5, lines 75–77) 

2. The authors grouped patients into strong cough group and weak cough group by use of cough score, which was the original score defined by the authors and assessed by bronchoscopists. The strong cough group included patients whose cough score was 0 or 1, and the weak cough group 2 or 3. However, the difference between cough score 1 (slight cough) and 2 (moderate cough, no need to interrupt procedure) was ill defined. The authors defined the cough score as objective score, but it depended on bronchoscopists’ subjectivity.

Response: Thank you for your comment. We revised the term from “subjective score” to “patient-based evaluation” as well as from “objective score” to “physician-based evaluation.”

3. The present study reported examination time was one of the factors which affected severe cough. Additionally, it is described that the procedure time in EBUS-TBNA groups was significantly longer than that in non-EBUS-TBNA-groups. The EBUS-TBNA groups included group 10-14, but they did not include group 2 (EBUS-TBNA). Why did the authors sort group 2 into the non-EBUS-TBNA-group? It was likely that the examination time did not be depend on EBUS-TBNA, but depended on using two procedures.

Response: Thank you for your important comments. We failed to show group 4 as a black column. The statistical analysis of examination time for the presence or absence of EBUS-TBNA in the original article was performed, including group 2. Thereby, we revised Figure 3A and demonstrated precisely the EBUS-TBNA group.

Minor points

1. The present study showed that patients in comfort group received more midazolam than those in discomfort group. The deep sedation could be important to comfort during bronchoscopy. In this study, there were patients who did not receive pethidine. How did the authors decide which they administered pethidine patients?

Response: Thank you for your important comments. 

Two patients who underwent bronchoscopy with only midazolam were on maintenance hemodialysis. We prefer midazolam to pethidine in cases with chronic renal failure. 

Therefore, we added the sentences in the sentences in the section of the “bronchoscopy procedure” section. 

The initial use and additional dosages of midazolam and pethidine were at the discretion of the primary bronchoscopist. Some patients on dialysis did not receive pethidine. (Page 7, line 105-107).

2. The authors described that patients’ discomfort reported to be associated with bronchoscopists’ experience in the discussion session. In this study, bronchoscopy was performed by pulmonologists whose experience ranged from 1 to 18 years. How did bronchoscopists’ experience affect the outcome of this study?

Response: 

Thank you for your important comments. As you pointed out, we re-analyzed the data for the bronchoscopists. In the first edition, we recorded the youngest physician when more than one bronchoscopist performed the bronchoscopy, but we decided to use the data of the bronchoscopist who performed the procedure the longest. Therefore, the results of the analysis for discomfort were highly variable, and both univariate and multivariate analyses showed that the experience of bronchoscopists had a significant impact on the patients’ discomfort. This means less-experienced bronchoscopists (less than 3 years) were identified as additional predictive factors for discomfort.

On the other hand, the bronchoscopist’s experience did not affect the cough during bronchoscopy.

We further added the sentences in the material and methods section as follows: “Collection of the associated data”; for the cases which were handled by more than one bronchoscopist. 

Among all enrolled cases, more than half of them underwent bronchoscopy by two or three bronchoscopists. In such cases, the bronchoscopist experience was recorded as the one who mainly performed the procedure. (Page 8, lines 129-131)

Reviewer #3: This study is very interesting. However some of detail is lacked.

First, how did you measured procedural time ? Is it from vocal code passing ? Is it to the end of procedure ? Please show me how.

Response: Thank you for your important comments. We added these sentences in the section of “Collection of the associated data”

bronchoscopy examination time; the time from insertion of the tip of the bronchoscope into the oral cavity to remove from the mouth (page 8, lines 120–121)

Second, the procedural time is too long comparing with standardized procedure, especially for EBUS-TBNA group, and this result may be a confounding factor.

Response: Thank you for your important comments. Exactly, EBUS-TBNA requires much time (median: 39 minutes), even if it is performed as a sole procedure (Fig 3A, group 2). Moreover, previous reports described that EBUS-TBNA needs 29 (PMID: 28967732) to 30 minutes (PMID: 28435314). In addition, our results suggest that EBUS-TBNA, examination time, and female sex were associated with strong cough independently on multivariate analysis. 

Third, Please tell me the use or not use of your additional lidocaine spraying. If not, you can reduce the cough level during procedure. And you'd better show the average dose of Midazolam and pethidine. Previously published report show that the usage of sedative drugs reduce the cough during BFS procedure. You should discuss about the appropriation of your sedation.

Response: Thank you for your comments. Regarding lidocaine spraying, we described the following: 

When the bronchoscope was passed through the larynx and trachea or coughing occurred during the procedure, 2% lidocaine was administered through the bronchoscopy channel. (Page 7, lines 108-109). 

For midazolam and pethidine, the total doses of midazolam and pethidine are shown in Table 3. The total dosages of midazolam and pethidine between discomfort and comfort were comparable. However, the total dosage of midazolam in the strong cough group was significantly higher than that in the weak cough group (Table 6), which seemed not to be the cause but the result of the strong cough.

We would like to thank the reviewers for their time and valuable comments, which have resulted in a significantly improved manuscript. We feel that the revised manuscript reflects suitable responses to the reviewers' comments, and we hope that revisions are completed to the Editor's satisfaction and it is now deemed suitable for publication in your journal.

Thank you for your consideration. We look forward to hearing from you.

Sincerely,

Takeshi Saraya MD, PhD

Department of Respiratory Medicine, Kyorin University School of Medicine, Mitaka, Tokyo, Japan

Address: 6-20-2, Shinkawa, Kyorin University School of Medicine

Post code:181-8611

City: Mitaka

Country: Japan

e-mail: sara@yd5.so-net.ne.jp

---

## [Decision Letter · Decision Letter 1]

8 Sep 2020

PONE-D-20-07274R1

Novel predictive factors for patient discomfort and severe cough during bronchoscopy: a prospective questionnaire analysis

PLOS ONE

Dear Dr. Saraya,

Thank you for submitting your manuscript to PLOS ONE. After careful consideration, we feel that it has merit but does not fully meet PLOS ONE’s publication criteria as it currently stands. Therefore, we invite you to submit a revised version of the manuscript that addresses the points raised during the review process.

ACADEMIC EDITOR:

　According to the reviewers' new comments, please refine the manuscript before publication.

We look forward to receiving your revised manuscript.

Kind regards,

Atsushi Miyamoto

Academic Editor

PLOS ONE

Reviewers' comments:

Reviewer's Responses to Questions

**Comments to the Author**

1. If the authors have adequately addressed your comments raised in a previous round of review and you feel that this manuscript is now acceptable for publication, you may indicate that here to bypass the “Comments to the Author” section, enter your conflict of interest statement in the “Confidential to Editor” section, and submit your "Accept" recommendation.

Reviewer #1: All comments have been addressed

Reviewer #2: (No Response)

Reviewer #3: All comments have been addressed

2. Is the manuscript technically sound, and do the data support the conclusions?

Reviewer #1: Yes

Reviewer #2: Partly

Reviewer #3: Yes

3. Has the statistical analysis been performed appropriately and rigorously? 

Reviewer #1: Yes

Reviewer #2: Yes

Reviewer #3: Yes

4. Have the authors made all data underlying the findings in their manuscript fully available?

Reviewer #1: Yes

Reviewer #2: Yes

Reviewer #3: Yes

5. Is the manuscript presented in an intelligible fashion and written in standard English?

Reviewer #1: Yes

Reviewer #2: Yes

Reviewer #3: Yes

6. Review Comments to the Author

Reviewer #1: The authors have addressed the reviewers' comments in this revision. I only have one more comment. Please update the p-values in Tables 3-7 with more accurate p-values rather than a range, e.g., <0.05.

Reviewer #2: The manuscript has been improved by the authors.

However, the problem has been remaining that the difference between cough score 1 (slight cough) and 2 (moderate cough, no need to interrupt procedure) was ill defined. How did the bronchoscopists decide which patients had slight cough or moderate cough?

Reviewer #3: (No Response)

7. PLOS authors have the option to publish the peer review history of their article (what does this mean?). If published, this will include your full peer review and any attached files.

Reviewer #1: No

Reviewer #2: No

Reviewer #3: No

---

## [Author Response · Author response to Decision Letter 1]

17 Sep 2020

Dear Reviewers

Thank you for your kindness to review our manuscript.

We revised the manuscript according to the reviewer's comments point by point.

We are very grateful to your great effort.

Thank you again.

Reviewer #1: The authors have addressed the reviewers' comments in this revision. I only have one more comment. Please update the p-values in Tables 3-7 with more accurate p-values rather than a range, e.g., <0.05.

R: Thank you for your comments. We revised them.

Reviewer #2: The manuscript has been improved by the authors.

However, the problem has been remaining that the difference between cough score 1 (slight cough) and 2 (moderate cough, no need to interrupt procedure) was ill defined. How did the bronchoscopists decide which patients had slight cough or moderate cough?

R: Thank you for your thoughtful comments. We revised them (Fig.2 and Table2) more precisely as follows.

Score Cough severity 

0 No cough 

1 A slight cough：go ahead the procedure

2 Moderate cough: transient interruption of the procedure in the trachea 

3 Severe cough: removal of the bronchoscope from the trachea

---

## [Decision Letter · Decision Letter 2]

22 Sep 2020

PONE-D-20-07274R2

Novel predictive factors for patient discomfort and severe cough during bronchoscopy: a prospective questionnaire analysis

PLOS ONE

Dear Dr. Saraya,

Thank you for submitting your manuscript to PLOS ONE. After careful consideration, we feel that it has merit but does not fully meet PLOS ONE’s publication criteria as it currently stands. Therefore, we invite you to submit a revised version of the manuscript that addresses the points raised during the review process.

ACADEMIC EDITOR:

Before acceptance, please confirm the inquiry from Reviewer1 as soon as possible.

We look forward to receiving your revised manuscript.

Kind regards,

Atsushi Miyamoto

Academic Editor

PLOS ONE

Reviewers' comments:

Reviewer's Responses to Questions

**Comments to the Author**

1. If the authors have adequately addressed your comments raised in a previous round of review and you feel that this manuscript is now acceptable for publication, you may indicate that here to bypass the “Comments to the Author” section, enter your conflict of interest statement in the “Confidential to Editor” section, and submit your "Accept" recommendation.

Reviewer #1: All comments have been addressed

Reviewer #2: All comments have been addressed

2. Is the manuscript technically sound, and do the data support the conclusions?

Reviewer #1: Yes

Reviewer #2: Yes

3. Has the statistical analysis been performed appropriately and rigorously? 

Reviewer #1: Yes

Reviewer #2: Yes

4. Have the authors made all data underlying the findings in their manuscript fully available?

Reviewer #1: Yes

Reviewer #2: Yes

5. Is the manuscript presented in an intelligible fashion and written in standard English?

Reviewer #1: Yes

Reviewer #2: Yes

6. Review Comments to the Author

Reviewer #1: In Table 3, there seems to be a typo. The numbers for Gender under Discomfort column should be 51/33 not 51/44.

Reviewer #2: (No Response)

7. PLOS authors have the option to publish the peer review history of their article (what does this mean?). If published, this will include your full peer review and any attached files.

Reviewer #1: No

Reviewer #2: No

---

## [Author Response · Author response to Decision Letter 2]

22 Sep 2020

Dear Reviewer1

Thank you for your kindness to review our manuscript.

We revised the Table3 as follows.

Thank you for your comments. 

We are very grateful to your great effort.

Thank you again.

---

## [Decision Letter · Decision Letter 3]

28 Sep 2020

Novel predictive factors for patient discomfort and severe cough during bronchoscopy: a prospective questionnaire analysis

PONE-D-20-07274R3

Dear Dr. Saraya,

We’re pleased to inform you that your manuscript has been judged scientifically suitable for publication and will be formally accepted for publication once it meets all outstanding technical requirements.

Kind regards,

Atsushi Miyamoto

Academic Editor

PLOS ONE

Additional Editor Comments (optional):

Reviewers' comments:

Reviewer's Responses to Questions

**Comments to the Author**

1. If the authors have adequately addressed your comments raised in a previous round of review and you feel that this manuscript is now acceptable for publication, you may indicate that here to bypass the “Comments to the Author” section, enter your conflict of interest statement in the “Confidential to Editor” section, and submit your "Accept" recommendation.

Reviewer #1: (No Response)

2. Is the manuscript technically sound, and do the data support the conclusions?

Reviewer #1: (No Response)

3. Has the statistical analysis been performed appropriately and rigorously? 

Reviewer #1: (No Response)

4. Have the authors made all data underlying the findings in their manuscript fully available?

Reviewer #1: (No Response)

5. Is the manuscript presented in an intelligible fashion and written in standard English?

Reviewer #1: (No Response)

6. Review Comments to the Author

Reviewer #1: (No Response)

7. PLOS authors have the option to publish the peer review history of their article (what does this mean?). If published, this will include your full peer review and any attached files.

Reviewer #1: No

---

## [Editor Report · Acceptance letter]

6 Oct 2020

PONE-D-20-07274R3 

Novel predictive factors for patient discomfort and severe cough during bronchoscopy: a prospective questionnaire analysis 

Dear Dr. Saraya:

I'm pleased to inform you that your manuscript has been deemed suitable for publication in PLOS ONE. Congratulations! Your manuscript is now with our production department. 

Kind regards, 

on behalf of

Dr. Atsushi Miyamoto 

Academic Editor

PLOS ONE